# A Potential Health Risk to Occupational User from Exposure to Biocidal Active Chemicals

**DOI:** 10.3390/ijerph17238770

**Published:** 2020-11-25

**Authors:** Joo-Hyon Kim, Moon-Young Hwang, Yoo-jin Kim

**Affiliations:** Division of Chemical Research, National Institute of Environmental Research, Hwangyeong-ro 42, Seo-gu, Incheon 22689, Korea; myrang@korea.kr (M.-Y.H.); yoojinolkim@korea.kr (Y.-j.K.)

**Keywords:** non-human hygiene disinfectant, insecticide, hazardous ingredients, exposure assessment, toxicological endpoint

## Abstract

Biocidal active chemicals have potential health risks associated with exposure to retail biocide products such as disinfectants for COVID-19. Reliable exposure assessment was investigated to understand the exposure pattern of biocidal products used by occupational workers in their place of occupation, multi-use facilities, and general facilities. The interview–survey approach was taken to obtain the database about several subcategories of twelve occupational groups, the use pattern, and the exposure information of non-human hygiene disinfectant and insecticide products in workplaces. Furthermore, we investigated valuable exposure factors, e.g., the patterns of use, exposure routes, and quantifying potential hazardous chemical intake, on biocidal active ingredients. We focused on biocidal active-ingredient exposure from products used by twelve occupational worker groups. The 685 non-human hygiene disinfectants and 763 insecticides identified contained 152 and 97 different active-ingredient chemicals, respectively. The toxicity values and clinical health effects of total twelve ingredient chemicals were determined through a brief overview of toxicity studies aimed at estimating human health risks. To estimate actual exposure amounts divided by twelve occupational groups, the time spent to apply the products was investigated from the beginning to end of the product use. This study investigated the exposure assessment of occupational exposure to biocidal products used in workplaces, multi-use facilities, and general facilities. Furthermore, this study provides valuable information on occupational exposure that may be useful to conduct accurate exposure assessment and to manage products used for quarantine in general facilities.

## 1. Introduction

Retail biocidal products are commonly used in and around residential places (public consumer), workplaces (occupational consumer), and multi-use facilities (general public). Because of the recent coronavirus 2 (SARS-CoV-2) pandemic, various types of biocidal products including various biocidal active-ingredient chemicals are being used worldwide for surface disinfection (non-human hygiene disinfection) [1]. For cleaning and disinfection of coronavirus 2, the Centers for Disease Control and Prevention (CDC) recommends cleaning and disinfecting high-touch surfaces at least once daily, assuming one had contact with the outside world in some way, i.e., either a person leaving and returning or goods coming in [2]. These products were readily accessible to the general population in retail markets and were intended to be used in contact with the external parts of the human body such as the epidermis or respiratory organs. A biocidal product means any active-ingredient chemical or mixture and is necessary to control harmful micro-organism or insects for the prevention of infection that can cause local or systemic effects raising several important considerations concerning public safety [3,4,5,6]. These products must be safe in their application, with a sufficient margin of safety to prevent any risk of adverse effects to the user [7,8]. Therefore, it is necessary to estimate exposure and health risk to users from the use of biocidal products. Exposure to biocidal products may result from inhalation and dermal contact during residential application. There are many inhalable formulations of biocidal products available to public and occupational consumers, including aerosol spray, trigger spray, electric liquid vaporizer, and mosquito coil. Occupational consumer exposures to biocidal products occurring in the workplaces may be direct. The magnitude of exposure for occupational workers to biocidal products is a function of exposure frequency and exposure amount. On the basis of occupational characteristics, the exposure amount of active-ingredient chemicals via consumer products varied for different occupation groups.

The Korean government enacted “The Korean Biocidal Products Regulation (KBPR)”, which concerns the placing on the market and use of biocidal products, which are used to protect humans, materials, or articles against harmful organisms, by the action of the active chemicals contained in the biocidal product. According to this regulation, all biocidal products should undergo exposure and risk assessments to evaluate health and environmental hazards caused by their use. The exposure assessment component of a risk assessment of hazardous chemicals requires evaluation of exposure of relevant chemicals through all the relevant pathways by all the relevant routes of exposure for all relevant periods. The products of exposure assessment are estimates of exposure of defined subjects to each chemical by periods and exposure pathways [9,10,11,12,13,14]. This regulation categorized users of biocidal products into non-professional consumer (public consumer and occupational consumer) and professional user (quarantine worker for public facilities and multi-use facilities). On the basis of this regulation, biocidal products were classified to non-human hygiene disinfectant, algaecide, insecticide, repellent, preservatives, and anti-fouling agent. This study took into account the exposure for occupational consumers using non-human hygiene disinfectants and insecticide products, which are available to these consumers for workplace use. Biocidal products are commonly used in several workplaces including multi-use facilities in Korea. Secondary exposure to the general public using multi-use facilities occurs through biocidal products used by occupational consumers. Human hygiene (directly applied to human) disinfectants regulated by the KFDA (Korean Food and Drug Administration) were excluded. The purpose of this study was to develop a representative database on exposure factors for biocidal products by occupational consumers and to evaluate toxicity characteristics of active-ingredient chemicals used in these products. These exposure and toxicity information data created in this study will be useful in establishing improved safety guidelines for biocidal products, conducting accurate assessments of consumer exposure, and better assessing risk to human health.

## 2. Materials and Methods

### 2.1. Interview–Survey Study

The consumers were categorized as general public consumers (household users) and occupational consumers. Occupational consumers were defined as consumers from occupational groups who use biocidal products in workplaces and public, multi-use facilities with high frequency because of their occupational characteristics. In order to characterize occupational consumers using biocidal product, the Korean Standard Classification of Occupation was referenced. Due to the characteristics of the occupation, the twelve occupational groups using a lot of biocidal products were selected. An interview–survey study was conducted to elucidate which biocidal products were commonly used in occupational places (workplaces) across all cities and provinces in Korea by occupational consumers. The interview–survey was carried out by a Korean survey company (K-STAT Research Ltd., Seoul, Korea) that we hired. The survey company has participants from all provinces, cities in Korea. Participants were selected as follows: (1) workers having occupation belonging to the twelve occupational groups and (2) workers using biocidal products in their workplace because of occupational characteristics. Among the participants who agreed to take the interview–survey, those who had experience using target biocidal products were subsequently individually interviewed. On the basis of the interview results, the survey company conducted a market search to identify and characterize the retail biocidal products and ingredient chemicals that participants used.

### 2.2. Interview–Survey Questionnaire

The interview–survey questionnaires consisted of information on biocidal products used in occupational places and the frequency of use, estimation of quantitative duration of exposure to products, quantitative amount of products used, and information on occupation (Appendix A). The active ingredients in biocidal products are generally listed with all other ingredients on the product label to ensure appropriate product use by consumers. If the active ingredients were not listed or lacked sufficient detail, Appendix A was obtained from the manufacturers and importers through the Korean National Institute of Environmental Research (NIER).

### 2.3. Use Pattern of Biocidal Products

Use information of biocidal products in occupational places was investigated to obtain the frequency of using products, the exposed duration of products, and the exposed amount per application by considering target organisms and the purposes of use. In order to investigate the exposed amount of biocidal products for occupational consumers, we purchased the various surveyed products from the market. The survey questionnaire included the used amount per products’ application. On the basis of the survey results, the exposure amount of products was investigated for occupational consumers. Experiments were conducted to evaluate an accurate amount of product used per application. The amounts of products used (g/use) were measured by weighing used amounts; the weight of the product was measured before and after use. We followed procedures recommended by the National Institute for Public Health and the Environment (RIVM) in the Netherlands [15].

### 2.4. Biocidal Products and Ingredients Survey Study

In order to elucidate the manufactured and imported tonnage of biocidal chemicals per year in Korea, the retail product types including these chemicals, and the amount of chemicals used as active ingredients in retail products, the survey company conducted an extensive phone and online survey. A total of 459 manufacturing and importing companies took part in this survey study. The survey questionnaire asked participant companies to list the active biocidal chemicals, manufactured and imported tonnage per year, and the biocidal products they produced and imported. Additionally, the mixing ratio of biocidal chemicals in products as an active ingredient was obtained from the participant companies.

### 2.5. Toxicological Information for Biocidal Active-Ingredient Chemicals

The toxicological health effects and reference toxicity values (i.e., chronic no observed adverse effect concentrations (NOAECs) and no observed adverse effect levels (NOAELs)) of active-ingredient chemicals were investigated. An occupational consumer exposure assessment was carried out according to the guidance from the information requirement and chemical safety assessment, which was described as an efficient, step-wise, and iterative procedure (e.g., characterize the substance, determine the scope of exposure assessment, build/retrieve the contributing use scenario, estimate the event exposure, and carry out risk characterization). The target routes of exposure were considered to be inhalation and dermal route according to usage purpose and application types of products. An evaluation of the toxicological data was carried out in relation to the respiratory and irritant effects of long-term exposure to the ingredients under investigation (European Chemicals Agency (ECHA), 2016). Official toxicology reports and studies (i.e., United State Environmental Protection Agent (U.S. EPA) documents, United State California Environmental Protection Agent (U.S. California EPA) documents, United State Registration Eligibility Decision (U.S. RED) report, and the European Union European Chemical Agency (EU ECHA) Dossier) were used for each chemical in order to investigate its toxicity value in the various products used. According to the toxicity value and health effects, dose rate, and toxicokinetic information should be considered. In particular, we derived reference toxicity values based on chronic NOAELs (no observed adverse effect levels). If the value of chronic NOAELs could not be derived, differences in toxicological values (i.e., LOAELs (lowest observed adverse effect levels) and NOAELs) were taken into consideration.

### 2.6. Statistical Analysis

All statistical analyses were conducted using SPSS version 22.0 (IBM, Armonk, NY, USA) differences with a *p* value of less than 0.05 were considered to be statistically significant unless noted otherwise. As a statistical analysis method to verify the difference between groups, T-test or one-way ANOVA was carried out.

## 3. Results

### 3.1. Characteristics of Occupational Consumer

For the purpose of exposure assessment, the different types of actual users need to be considered. As a first step, depending on the potential use of biocidal products, non-professional users (consumers) were categorized into public consumers (user in house) and occupational consumers (user in occupational place) who are likely to be exposed directly to the retail biocidal products. This study focused on active-ingredient exposure from biocidal products used by occupational workers in their place of occupation, multi-use facilities, and general facilities. As a result of market searching, nine predominant usage purposes of non-human hygiene disinfectant and three categories of insecticide for common target insects (e.g., mosquitoes, cockroaches, houseflies) included for elimination or control were found in Korean retail markets. On the basis of the Korean Standard Classification of Occupation, we estimated that workers in the twelve occupational groups (e.g., livestock occupation, medical-service-related business, and health service business) could come into contact with biocidal products as a consequence of their occupational life (Table 1).

As a second step, the interview–survey approach obtained the database about several subcategories of the twelve occupational groups, biocidal product use pattern in occupational places, and exposure information of two biocidal product groups. A total of 2432 respondents, who had occupations belonging to the twelve occupational groups, completed the formal questionnaire during the interview–survey. Participants were continuously contacted for the interview survey until the required number of respondents (over 2000) was reached. On the basis of the survey results, it was determined that the survey respondents were using various biocidal products in occupational places and were regularly using them. The suboccupational distribution results of 2432 respondents (e.g., livestock occupation-livestock breeder, dairy worker, and livestock-industry-related machinery operation employee) are summarized in Table 2.

### 3.2. Exposure Factor Values for Occupational Consumers

The occupational use frequencies of biocidal products for total respondents were investigated. Survey responses were disaggregated based on occupational categories. Table 3 summarizes the mean and standard deviation (s.d.) of product use frequency and use amount values for each of the occupational consumer groups. The occupational consumer directly applying biocidal products in the workplace experiences primary exposure during application and secondary exposure after application of the product. The general public using these public-use facilities (e.g., medical-service-related business, companion animal business, children care business, and health service business) unintentionally experience secondary exposure. Based on the survey results, the frequency of biocidal product use was varied by product purpose, target insects, and occupations. In the case of insecticides used to eliminate or control mosquitoes, cockroaches, and houseflies, the use frequency of housefly and mosquito insecticide products used by fisheries, food cooking/sales business, and pet beauty/care business occupational consumers were relatively higher than that of other occupational consumer groups with mean use frequencies ranging from 1.2 to 2.7 uses/day. Compared to the use frequencies of mosquito insecticides and housefly insecticides, respondents showed little use of cockroach insecticide products (Table 3). Comparatively, to disinfect microorganisms, multi-purpose disinfectants use by veterinarians (companion animal business) had the highest use frequency. In order to evaluate the biocidal product use amount of occupational consumer groups, we compared the difference in the weight of the product before and after use at room temperature. The use amount of housefly insecticide products used by livestock occupational consumers was the higher than that of other occupational consumer groups. The mean use amount of agriculture/forestry occupational, fishery occupational, and pet beauty and care occupational consumers for housefly insecticides was comparatively higher than that of other occupational consumers. These results implied that respondents using products were exposed to active-ingredient chemicals in biocidal products.

### 3.3. Exposed Amount for Occupational Consumers

Considering to the worst-case scenario about exposure amounts, we summed up the use amount of biocidal product groups divided by insecticide and non-human hygiene disinfectant per day and calculated the exposure amount of product groups according to use frequency and use amounts (Table 4). The exposed amounts of respondents to biocidal products per day implied the exposure of respondents to the combined/cumulative exposed amounts of active-ingredient chemicals per day. In the case of occupational consumers engaged in the fishery occupation, the pet beauty and care business, and the beauty/lodging/facility management business, the mean exposed amount per day was determined to be relatively high 131.7 g/day (insecticide), 125.1 g/day (insecticide), and 124.1 g/day (disinfectant). In addition, the exposure amount per day of total biocidal products to occupational consumers engaged in the agriculture/forestry occupation, the fishery occupation, the livestock occupation, the food cooking and sale business, and the beauty/lodging/facility management business were relatively high. Combined use of biocidal products that contain several (or same) active-ingredient chemicals results in a cumulative exposure to those chemicals.

### 3.4. Identification of Active-Ingredient Chemicals in Biocidal Products

The search for biocidal products available in retail markets identified 685 non-human hygiene disinfectants and 763 insecticides that offered substantial diversity in product purpose and active-ingredient chemicals. In addition, we categorized a subset of searched products, dividing them by the purpose for non-human hygiene disinfectants and by the target insect intended for elimination or control for a more in-depth analysis. Based on the results of the interview–survey to the manufacturers and importers, 152 active-ingredient chemicals used in non-human hygiene disinfectants, 97 active-ingredient insecticides, and the mixing ratio of these chemicals in products were identified. The purpose for insecticides was often defined based on the specific target insect intended for elimination or control. Common target insects included mosquitoes, cockroaches, and houseflies. Furthermore, in order to estimate potential exposure by grasping the amount of active-ingredient chemicals distributed in South Korea, a total tonnage of manufactured and imported biocidal products per year in Korea was surveyed (Table 5).

The 685 non-human hygiene disinfectants and 763 insecticides identified in the product survey contained 152 and 97 different active-ingredient chemicals, including sodium hypochlorite, ethanol, hypochlorous acid, hydrogen peroxide, and didecyldimethylammonium chloride in disinfectants and tetramethrin, d-phenothrin, deltamethrin, and hydramethylnon in insecticides. Among the 685 disinfectants included in the subset, 108 products contained sodium hypochlorite and 83 products contained ethanol as the predominant active ingredient. Based on the results of the distributed number of retail products and total tonnage per year of manufactured and imported ingredients, several ingredient chemicals such as sodium hypochlorite, chlorine dioxide, and temephos were used in retail products in South Korean market. Table 6 summarizes the ten most common active ingredients used in non-human hygiene disinfectants and insecticides, including their prevalence, maximum concentrations, and total tonnage of manufactured and imported biocidal products per year. The surveyed mixing ratio of active-ingredient chemicals (%) in the product groups are listed. Among the 152 substances, sodium hypochlorite was used as the active ingredient in 108 non-human hygiene disinfectants at a mixing ratio of 0.01–100%. Tetramethrin was used in 70 of the 763 insecticides. The maximum mixing ratio of sodium hypochlorite, ethanol, chlorine dioxide, λ-cyhalothrin, and zeta cypermethrin exceeded 90% in products.

Among the surveyed active-ingredient chemicals, the levels of mixing ratio in products, the distribution of retail products including these ingredients, vapor pressure, and the structure for ten major ingredient chemicals are summarized in Table 6. Sodium hypochlorite, hydrogen peroxide, and ethanol were used from low levels to high levels in retail non-human hygiene disinfectants. The mixing ratio of hypochlorous acid in retail products was relatively high, i.e., it was used at 50–100% in 34 products. In the case of chemicals used in insecticide products, the mixing ratio of the five chemicals was relatively low at below 20%. In the case of ingredient chemicals used in insecticides compared to chemicals in non-human hygiene disinfectants, tetramethrin, d-phenothrin, deltamethrin, and others were of relatively low vapor pressure.

### 3.5. Toxicological Endpoint for Active-Ingredient Chemicals in Biocidal Products

Toxicity values and clinical health effects of active-ingredient chemicals were determined through a brief overview of previous toxicity studies aimed at estimating human health risks, i.e., risks to product users and secondarily exposed by-standers (Table 7). The toxicity evaluation of active-ingredient chemicals by this study was based on the principles and practice of the risk assessment process usually applied for ingredients in retail products. Hazard identification is carried out to identify whether the chemical has the potential to damage human health. It is based on the results of in vivo tests, in vitro tests, clinical studies, accidents, and human epidemiological studies. Moreover, intrinsic physical, chemical, and toxicological properties of the molecule under consideration are taken into consideration. In dose–response assessment, the relationship between the toxic response and the exposure is studied. In the case of an effect with a threshold, the dosage at which no adverse effects are observed is determined. The toxicological data were assessed based on long-term exposure to the active ingredients and exposure routes. The U.S. EPA (United States Environmental Protection Agency), Cal/EPA (United States California Environmental Protection Agency), ECHA (European Chemicals Agency) registration dossiers, the OECD-generating profiles (the screening information dataset (SIDS) initial assessment profile), and KOSHA (Korean Occupational Safety and Health Research Institute) reports were used to evaluate the toxicity characteristics of active-ingredient chemicals. These reference toxicity values were used to identify the risks to human health of these chemicals. The product users’ and secondarily exposed by-standers’ exposure to biocidal products occurs through any or all of two potential exposure routes: inhalation and dermal contact. Inhalation is the predominant exposure route of these products. Among 20 active-ingredient chemicals, 9 chemicals had inhalation toxicity values. However, inhalation toxicity information for other chemicals was not found. According to the ECHA registration dossier of sodium hypochlorite, hypochlorous acid, and chlorine dioxide, these ingredient chemicals release active chlorine gas, e.g., active chlorine or available/releasable chlorine, which is a disinfectant, algaecide, and micro-biocide. The potential toxic effects of these chemicals is calculated as AEC_inhalation_ (external reference value for inhalation effects) of chlorine gas. Depending on chlorine concentrations, signs of toxicity ranged from dyspnea and coughing, irritation of the throat and eyes, headache, to temporary changes in lung function, cytopathological features, and tracheobronchial congestion. Exposure to 0.5 ppm (1.5 mg/m^3^) chlorine gas resulted in only trivial changes of lung-function parameters, therefore the NOAEC (no observed adverse effect concentration) was derived at 1.5 mg/m^3^.

## 4. Discussion

This study aimed to create a national exposure factor database for use in exposure and risk assessments of biocidal products in terms of human health. The study mainly showed that biocidal product use in prevalent occupational workplace, multi-use facilities, and general facilities could cause potential exposure of occupational consumers and the general public to their active-ingredient chemicals. Primary exposure to active-ingredient chemicals occurs to the consumer who actively uses the biocidal products. Secondary exposure is exposure that may occur after the actual use or application of the product. Primary exposures are invariably higher than secondary exposures [16]. Among workplaces of occupational consumers, food cooking and food sales business, medical-service-related business, companion animal business, children care business, health service business, and beauty/lodging/facility management business are related to the public and multi-use facilities. Furthermore, products used by occupational consumers working in the medical-service-related business and child care business might affect the health risk of preschool children and children.

In order to estimate the health risk of active-ingredient chemicals to occupational consumers, the health risk study was carried out in steps: ① information on occupation characteristics using retail biocidal products and occupational consumers, ② retail product purchase and use: list of biocidal products used, ingredient chemicals, the mixing ratios in products, ③ toxicity identification and characterization of ingredient chemicals, ④ exposure assessment (determining exposure factors to occupational consumers); the frequency of use; qualitative descriptions of product use habits [16,17,18,19].

The next step is health risk assessment, this assessment is ongoing as a further study. Fundamental to the health risk assessment process is the estimation of human exposure to the active-ingredient chemicals in retail biocidal products. The aim of the toxicity (hazard) identification is to identify the health effects of concern. Hazard characterization (dose–response assessment) is the estimation of the relationship between dose or level of exposure to ingredient chemical and the incidence and severity of an effect [16,18]. This study carried out toxicity identification and characterization of twelve ingredient chemicals used in retail non-human hygiene disinfectants and insecticides. To understand the pattern of retail biocidal product use by occupational consumers, exposure information of biocidal products via inhalation and dermal contact was obtained. The lack of product exposure information and exposure assessment was a major limitation of the health risk assessment study. The exposure assessment study consists of four steps; ① determine occupational consumers using biocidal products in workplaces, ② determine retail biocidal products and ingredient chemicals (maximum concentration/minimum concentration) in workplaces, ③ investigate exposure information including use frequency and use amount, and ④ calculate exposed amount per day of occupational consumers. The lack of a reliable exposure database for biocidal products was a major limitation of the risk assessment study [12,20].

The specific pattern of use data requirements for different biocidal products are purpose of product (physical properties, where used, description of tasks, target organisms), use environment (pattern of control, use pattern), loading phase (task, frequency per task, duration of task, quantity used per task), application phase, post-application phase, and others. The essential information of use pattern requires the derivation of exposure scenarios, which are then evaluated to derive quantitative exposure estimates [16,17]. For the survey used in this study, respondents had to declare the frequency, duration, and amount of product used per application of biocidal products, given several options (SI 1). However, surveys have their limitations. A lack of accurate memory regarding the questions posed might impact the quality of the data; in-person interview surveys might yield data of better quality [21,22]. For health risk assessments concerning volatile ingredients, vapor pressure of ingredient chemicals should be considered. In order to measure the accurate concentration of volatile ingredients in the air of occupational environment, residual volatile ingredients emitted into the indoor air should be measured. However, in this study, we could not measure these amounts and concentrations in indoor air. This approach might be useful in establishing guidelines of exposure assessment for occupational consumers. To protect consumers and public users from several hazardous ingredient chemicals, more comprehensive exposure estimation and assumption are needed.

## 5. Conclusions

This study investigated a fundamental approach to assess the occupational exposure to biocidal active ingredients by using products in occupational places (workplaces), multi-use facilities, and general facilities. The process of assessing exposure to biocidal products used in workplaces requires determining the patterns of use (exposure factors), identifying the exposure population (occupational consumers), establishing exposure routes (inhalation and dermal exposure), and quantifying potential ingredient chemicals intake. This study determined the recent exposure factors using an interview survey of over 2400 occupational consumers using biocidal products in the occupational place, multi-use facilities, and general facilities. Estimating occupational exposure to biocidal active-ingredient chemicals via using products is a fundamental element of the health risk assessment process. Furthermore, we calculated the exposure amount of biocidal product used by occupational consumers. The exposure amount of products could be used for estimating the exposure amount to their ingredient chemicals of occupational consumers. Additionally, toxicological characteristics of ingredient chemicals were evaluated considering the characteristics of occupational characteristics and biocidal products categories. As a further study, the health risk assessment study of active ingredients to occupational consumers and public users of multi-use facilities and general facilities as a second exposure estimation were processed.

## Figures and Tables

**Table 1 ijerph-17-08770-t001:** Categories of studied biocidal product and product consumers.

Biocidal Products Categories	Consumers
Non-human hygiene disinfectant For kitchen For bathroom For fungi in bathroom For drainage For toilet For air conditioner For children’s goods For companion animals For multi-purpose Insecticides (target organism) Mosquitoes Cockroaches Houseflies	Public consumer Household user Occupational consumer (Categories of occupation) Agriculture occupation/Forestry occupation Fisheries occupation Livestock occupation Food cooking and food sales business Medical service related business Fabric production and sales business Funeral business Companion animal business Children care business Health service business Beauty/lodging/facility management business

**Table 2 ijerph-17-08770-t002:** Categories of occupational consumers using biocidal products due to occupation.

Categories of Occupational Consumers
Occupational consumer divided by occupation characteristics Agriculture occupation/ Forestry occupation Landscape worker Forestry-related employee Crop plantation worker Agricultural machinery operation employee Fisheries occupation Fishery Fishing-related machinery operation employee Livestock occupation Livestock breeder Dairy worker Livestock-industry-related machinery operation employee Food cooking and food sales business Food making and processing related employee Food making machinery operation employee Food service worker Chef and cook Medical-service-related business Medical-service-related worker Fabric production and sales business Textile and leather processing worker Laundry-related worker Ironing-related worker Funeral business Funeral counselor and funeral worker Companion animal business Veterinarian Pet beauty and care-related worker Children care business Child care service worker Children’s goods rental business worker Child daycare teacher Kindergarten teacher Health service business Beauty/lodging/facility management business Skin care service worker Accommodation staff Facility management worker Facility cleaner

**Table 3 ijerph-17-08770-t003:** Exposure factor values of biocidal products adjusted for occupational consumers.

Occupational Consumers		Biocidal Products Categories (Summer Season Use)		Use Frequency (use/day)		Use Amount (g/use)	
				Mean	S.D.	Mean	S.D.
Agriculture/ Forestry occupation (*n* = 312)		Insecticides	Mosquitoes	1.8	1.6	17.6	18.0
			Cockroaches	0.1	0.2	23.3	17.9
			Houseflies	1.4	1.3	45.2	43.1
Fisheries occupation (*n* = 149)		Insecticides	Mosquitoes	1.8	1.5	17.3	18.4
			Cockroaches	0.2	0.3	19.1	7.8
			Houseflies	2.1	2.0	43.9	49.3
Livestock occupation (*n* = 298)		Insecticides	Mosquitoes	1.2	1.4	19.3	24.3
			Cockroaches	0.1	0.2	29.5	20.9
			Houseflies	1.0	1.2	56.6	73.5
Food cooking and sales business (*n* = 464)		Disinfectants	Kitchen	0.5	0.5	54.9	81.3
			Remove fungi	0.2	0.3	32.8	37.1
			Drainage	0.2	0.2	103.8	192.4
			Toilet	0.2	0.2	42.7	54.4
			Multi-purpose	0.7	1.2	41.5	57.4
			Air conditioner	0.04	0.08	31.6	38.7
		Insecticides	Mosquitoes	2.3	2.5	16.7	15.9
			Cockroaches	0.05	0.09	25.8	18.1
			Houseflies	2.7	2.5	19.8	30.3
Medical-service-related business (*n* = 80)		Disinfectants	Multi-purpose	0.9	1.1	41.3	87.1
		Insecticides	Mosquitoes	1.4	1.2	9.3	13.5
			Cockroaches	0.1	0.2	20.0	6.3
			Houseflies	1.6	1.3	21.9	18.4
Fabric production and sales business (*n* = 299)		Insecticides	Houseflies	0.1	0.2	13.4	16.4
Funeral business (*n* = 21)		Disinfectants	Multi-purpose	0.4	0.3	49.3	67.6
		Insecticides	Mosquitoes	1.8	0.9	18.6	17.2
			Houseflies	0.2	0.2	18.3	13.9
Companion animal business (*n* = 235)	Veterinarian	Disinfectants	Multi-purpose	1.0	1.9	36.3	54.9
		Insecticides	Mosquitoes	1.4	1.1	13.6	14.9
			Cockroaches	0.1	0.1	13.8	6.0
			Houseflies	1.1	0.8	28.9	27.2
	Pet beauty and care	Disinfectants	Multi-purpose	0.8	0.8	33.0	30.4
			Animal care	0.6	0.6	18.3	31.6
		Insecticides	Mosquitoes	1.2	1.1	13.5	15.4
			Cockroaches	0.1	0.1	21.4	15.0
			Houseflies	2.4	0.4	44.1	59.8
Children care business (*n* = 200)		Disinfectants	Multi-purpose	0.9	2.4	33.3	48.3
			Children goods	0.6	2.1	14.4	26.9
		Insecticides	Mosquitoes	1.5	1.7	5.4	11.5
			Cockroaches	0.05	0.06	20.1	12.3
			Houseflies	0.6	0.3	11.5	10.3
Health service business (*n* = 24)		Disinfectants	Multi-purpose	0.9	1.1	41.3	87.1
		Insecticides	Mosquitoes	1.4	1.2	9.3	13.5
			Cockroaches	0.1	0.2	20.0	6.3
			Houseflies	1.6	1.3	21.9	18.4
Beauty/lodging/facility management business (*n* = 350)		Disinfectants	Kitchen	0.5	0.5	54.7	81.2
			Remove fungi	0.2	0.2	37.0	42.3
			Drainage	0.2	0.2	107.9	185.8
			Toilet	0.2	0.2	45.2	62.1
			Multi-purpose	0.8	1.4	46.7	97.7
			Air conditioner	0.08	0.1	24.6	33.2
			Children goods	0.6	2.1	14.4	26.9
			Animal care	0.6	0.6	18.3	31.6
		Insecticides	Mosquitoes	1.7	1.7	15.9	18.8
			Cockroaches	0.09	0.1	25.1	18.2
			Houseflies	1.5	1.7	43.7	56.9

**Table 4 ijerph-17-08770-t004:** Use amount and exposure amount of products used by occupational consumers.

Occupational Consumers		Biocidal Products	Use Amount (g/use)		Exposure Amount per Day (g/day) ^c^	
			Mean	S.D.	Mean	S.D.
Agriculture/ Forestry occupation		Insecticides ^a^	86.2	79.0	99.5	92.7
Fishery occupation		Insecticides ^a^	80.3	75.5	131.7	133.8
Livestock occupation		Insecticides ^a^	105.4	118.8	82.7	96.5
Food cooking and sale business		Disinfectants ^b^	307.4	461.7	93.3	172.9
		Insecticides ^a^	62.4	64.5	93.3	117.4
Medical-service-related business		Disinfectants ^b^	41.3	87.1	37.3	95.8
		Insecticides ^a^	51.3	38.2	50.2	41.3
Fabric production and sale business		Insecticides ^a^	33.4	48.43	1.3	3.2
Funeral business		Disinfectants ^b^	49.3	67.6	19.7	20.2
		Insecticides ^a^	36.9	31.1	37.1	18.1
Companion animal business	Veterinarian	Disinfectants ^b^	87.7	117.0	36.3	104.4
		Insecticides ^a^	65.6	138.5	52.2	38.8
	Pet beauty and care	Disinfectants ^b^	51.3	62.0	37.4	43.3
		Insecticides ^a^	79.0	90.2	125.1	42.3
Children care business		Disinfectants ^b^	47.8	75.3	38.6	172.7
		Insecticides ^a^	37.1	34.2	16.0	23.4
Health service business		Disinfectants ^b^	41.3	87.1	37.2	95.8
		Insecticides ^a^	51.3	38.2	50.2	41.3
Beauty/lodging/facility management business		Disinfectants ^b^	349.2	561.2	124.1	314.2
		Insecticides ^a^	84.8	94.0	94.8	130.5

^a^ Insecticides (Mosquitoes + Cockroaches + House flies), ^b^ disinfectants (Kitchen + Remove fungi + Drainage + Toilet + Multi-purpose + Air conditioner), ^c^ exposed amount per day = use frequency(use/day) × use amount(g/use).

**Table 5 ijerph-17-08770-t005:** Retail products and their active-ingredients characteristics based on distribution and sale.

Biocidal Products and Ingredients	No. of Retail Products	Mixing Ratio in Product (%)		Total Tonnage (Production and Import/Year)
		Min. ^a^	Max. ^b^	
**685 non-human hygiene disinfectants (** **117,670 ton) and 152 ingredients (12,267 ton)**				
Sodium hypochlorite (chlorine-based) (NaOCl/NaClO/ClNaO)	108	0.01	100	6087 ton/year
Ethanol (C_2_H_6_O)	83	0.3	100	233 ton/year
Hypochlorous acid (HOCl/HClO)	39	0.00001	0.3	<10 ton/year
Hydrogen peroxide (H_2_O_2_)	26	0.05	70	242 ton/year
Didecyldimethylammonium chloride (C_22_H_48_ClN)	23	0.05	51	<10 ton/year
C12-18-Benzalkonium chloride (Alkyl(C=12-18) benzyl dimethyl ammonium chloride)	19	0.1	2.61	<10 ton/year
Chlorine dioxide (chlorine-based, ClO_2_)	19	0.01	99.5	396 ton/year
Quaternary ammonium compounds (C-12-16-Benzalkonium chloride)	18	0.212	55	<10 ton/year
Silver (Ag)	15	0.001	2.5	<10 ton/year
2-Propanol (isopropyl alcohol, C_3_H_8_O)	13	0.01	90	<10 ton/year
**763 Insecticides (21,458 ton) and 97 Ingredients (1312 ton)**				
Tetramethrin (phthalthrin, C_19_H_25_NO_4_)	70	0.1	1.07	26 ton/year
d-phenothrin (3-Phenoxybenzyl 2-dimethyl-3-(methylpropenyl)cyclopropanecarboxylate)	64	0.03	10	21 ton/year
Deltamethrin (decamethrin, C_22_H_19_Br_2_NO_3_)	59	0.05	3	16 ton/year
Hydramethylnon (C_25_H_24_F_6_N_4_)	45	0.3	2	<1 ton/year
Fipronil (C_12_H_4_Cl_2_F_6_N_4_OS)	42	0.003	11.7	103 ton/year
Diflubenzuron (C_14_H_9_ClFN_2_O_2_)	41	2	82.81	<1 ton/year
λ-Cyhalothrin (C_23_H_19_ClF_3_NO_3_)	34	1.5	97.5	<1 ton/year
Zeta cypermethrin (C_22_H_19_Cl_2_NO_3_)	34	0.09	95	8 ton/year
Permethrin (C_21_H_20_Cl_2_O_3_)	33	0.1	55	10 ton/year
Temephos (C_10_H_20_O_6_P_2_S_3_)	18	1.0	50	505 ton/year

^a^ Minimum concentration in product, ^b^ maximum concentration in product.

**Table 6 ijerph-17-08770-t006:** Mixing ratio of major chemicals used in products as ingredients.

Products\Chemicals	No. of Retail Products Including Ingredient Chemicals (Structure, Vapor Pressure (Pa))
Sodium Hypochlorite(1470–2000)	Ethanol(7870, 25 °C)	Hypochlorous Acid(0.933, 25 °C)	Hydrogen Peroxide(678.61–2813.1, 25 °C)	DDAC(0.944 × 10^−8^, 20 °C)
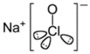	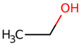	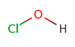	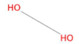	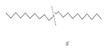
**Disinfectants**	<5%	51	6	2	9	12
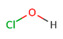	<10%	25	2		11	4
<20%	14	25		1	7
<30%	6	16		0	
<50%	5	16		2	
≤100%	7	18	37	3	
	**Tetramethrin** **(0.944 × 10^−3^, 30 °C)**	**d-phenothrin** **(0.16 × 10^−3^, 20 °C)**	**Deltamethrin** **(0.145 × 10^−4^, 25 °C)**	**Hydramethylnon** **(0.27 × 10^−5^, 25 °C)**	**Fipronil** **(0.399 × 10^−6^, 25 °C)**
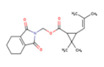	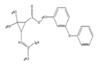	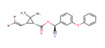	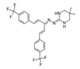	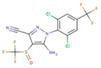
Insecticides	<5%	70	53	59	45	44
<10%		8			1
<20%		3			
<30%					
<50%					
≤100%					
Target insect	Mosquitoes ^a,b^Flies ^a^Cockroaches ^a,b^	Mosquitoes ^a^Flies ^a^Cockroaches ^a^	Mosquitoes ^a,b^Flies ^a^Cockroaches ^a^	Cockroaches ^a,b^Ants ^a^	Cockroaches ^a,b^Ants ^a^

^a^ Insecticidal effect, ^b^ larvicidal effect.

**Table 7 ijerph-17-08770-t007:** Summary of toxicological endpoint for ingredients in biocidal products.

Chemicals	Toxicity Value	Health Effects
Sodium hypochlorite	AEC_inhal_ = 0.5 mg/m^3^ (available chlorine, inhalation) ^a^	Total lung capacity was lower, pulmonary function was transiently affected
Ethanol	NOAEL = 2400 mg/kg/day (90 days/rat, oral) ^b^	Hepatic yellowing, centrilobular steatosis
Hypochlorous acid	AEC_inhal_ = 0.5 mg/m^3^ (available chlorine, inhalation) ^a^	Total lung capacity was lower, pulmonary function was transiently affected
Hydrogen peroxide	NOAEC = 81 mg/kg/day (90 days/mice, oral) ^b^	Duodenal mucosal hyperplasia
Didecyldimethylammonium chloride	LOAEL = 0.11 mg/m^3^ (13 weeks/rat, inhalation) ^f^	Infiltration of inflammatory cells, pneumonia interstitialis in lung and trachea
C12-18-Benzalkonium chloride	NOAEL = 0.22 mg/ m^3^ (13 weeks/rat, inhalation) ^e^	Histopathological changes in trachea, lung, and bronchial lymph nodes
Chlorine dioxide	AEC_inhal_ = 0.5 mg/m^3^ (available chlorine, inhalation) ^a^	Total lung capacity was lower, pulmonary function was transiently affected.
Quaternary ammonium compounds	LOAEL_ADJ_ = 0.11 mg/m^3^ (13 weeks/rat, inhalation) ^e^	Increased breathing frequency
Silver	NOAEL_ADJ_ = 0.02 mg/m^3^ (13 weeks/rat, inhalation) ^c^	Chronic alveolar inflammation, macrophage accumulation
2-Propanol	NOAEL_HEC_ = 210 mg/m^3^ (104 weeks/rat, inhalation) ^d^	Kidney lesion, fetal growth delay
Tetramethrin	LOAEL = 134 mg/m^3^ (90 days/rat, inhalation) ^c^	Urinalysis, hypertrophic, liver necropsy findings, hyaline droplets in the kidney
d-Phenothrin	NOAEL = 8.2 mg/kg/day (ADI = 0.08) (52 weeks/dog, oral) ^c^	Mild anemia
Deltamethrin	NOAEL = 1 mg/kg/day (104 weeks/rats, oral) ^a^	Neurological disorders (gait abnormalities)
Hydramethylnon	NOAEL = 1 mg/kg/day (chronic dog study, oral) ^e^	Increased incidence of soft stool, mucoid stool, and diarrhea observed
Fipronil	NOAEL = 2 mg/kg/day (repeated toxicity, oral) ^e^	Not specified
Diflubenzuron	RfD = 0.02 mg/kg/day (NOEL = 2 mg/kg/day, 52 weeks/dogs, oral) ^f^	Affects the hemoglobin of animals in studies
λ-Cyhalothrin	RfD, cPAD = 0.001 mg/kg/d (chronic dog study, oral) ^e^	Clinical signs of neurotoxicity (abnormal gait)
Zeta cypermethrin	NOAEL = 5 mg/kg/day (2 years/rat, oral) ^c^	Liver and kidney toxicity
Permethrin	NOEL = 5 mg/kg/day (ADI = 0.05) (1 year/rat, oral) ^c^	Histopathological changes in the adrenals, hepatic cellular swelling
Temephos	NOAEL = 0.3 mg/kg/day (90 days/rat, oral) ^e^	Red blood cell cholinesterase inhibition

AEC_inhal_—external reference value for inhalation local effects; NOAEC—no observed adverse effect concentration; NOAEL—no observed adverse effect level; NOEL—no observed effect level; LOAEC—lowest observable adverse effect concentration; cPAD—chronic population-adjusted dose; RfD—reference dose; ^a^ European Chemicals Agency (ECHA), BPC TOX-WG III-2016; ^b^ Organization for Economic Co-operation and Development (OECD), Screening Information Dataset (SIDS) report; ^c^ European Chemicals Agency (ECHA), registration dossier; ^d^ United States California Environmental Protection Agency (US California EPA); ^e^ United States Environmental Protection Agency (US EPA), Reregistration Eligibility Decision (RED) report; ^f^ Korean Occupational Safety and Health Research Institute (KOSHA), subchronic inhalation toxicity study of Didecyldimethylammonium chloride (DDAC) report.

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
