# Peer review of "A Potential Health Risk to Occupational User from Exposure to Biocidal Active Chemicals"

_ijerph, 2020, doi:10.3390/ijerph17238770_

Round 1

Reviewer 1 Report

This study investigated the biocidal products and active ingredient chemicals used by occupational consumers, examined the toxicological information and health effects of biocides, and calculated the exposure factor values.

As the crisis of COVID-19 continues, the health risk assessment of biocidal products, including a disinfectant, is becoming an important task, and therefore this manuscript is very timely to identify health risks for occupationally exposed workers using biocides. However, methods and results are mixed up. Some of the design choices are described as a result. Overall, it is difficult to understand what was done and why. I do have some comments for the authors to consider:

  1. Abstract: The entire manuscript needs to be refined in English. Some of the words are ambiguous, and the sentences are quite long. Also, the results and conclusions are not fully stated.
  2. 2.1. Interview-survey study: Did the author receive any Research Ethics Approval (IRB) from the Research Ethics Committee for interview surveys, online, and telephone surveys? Since the authors surveyed workers, this paper should receive ethics approval from the Research Ethics Committee.
  3. 2.4. Biocidal products and ingredients survey study: Why is it important to know how much biocidal substances are manufactured and imported in Korea? The authors simply state that the total tonnage in Table 5 only, and no interpretation of the results can be found in the text. If this investigation is necessary, the results will need to be interpreted. In order for the author to calculate the amount of biocidal substances circulating in Korea, a survey on the amount of exports is also required. According to the Korea national statistical survey of chemical substances, the amount used is calculated as manufactured volume + import volume - export volume. The authors should additionally calculate the export volume to know the distribution of chemicals.
  4. 2.4. Biocidal products and ingredients survey study: The biocidal products and ingredients used were investigated through the interview survey. However, why did the authors research the same questions (products and active ingredients) from 459 companies? Are the 459 companies included in all 12 occupational categories in the interview survey? To help readers understand, the authors need to write down the relationship between the interview survey and the online/telephone survey.
  5. 2.5. Toxicological information for biocidal active ingredient chemicals: Why the authors only investigated chronic toxicity information? Among the toxicity information, carcinogenicity, reproductive toxicity, and germ cell mutagenicity are particularly harmful substances. Have the authors considered investigating this toxicity information? If not, the author should consider adding the following toxicity information.

Author Response

Thank you very much for your comments and great suggestions. Your detailed and very valuable comments and recommendations improved the quality and presentation of this manuscript. I appreciate your time and effort. I answered to your questions and comments item by item. Most of your suggestions were reflected in the revision, and were detailed in the following.

Q1) Abstract: the entire manuscript needs to be refined English. Some of the words are ambiguous, and the sentence are quiet long. Also the results and discussions are not fully stated.

A1) Thank you for your comment. I revised abstract, according to your comments.

Abstract: Biocidal active chemicals have potential health risks associated from exposure to retail biocide products pesticides and disinfectants such as personal disinfectants for COVID-19. Reliable exposure assessment was investigated to understand the exposure pattern of biocidal product used by occupational workers in occupational place, multi-use facilities, and general facilities. Exposure to hazardous biocidal chemicals via products used by occupational workers in occupational place, multi-use facilities, and general facilities may play a role in health risk of occupational workers and general public caused by use of ingredients. The interview-survey approach was conducted to obtain the database about several sub-categories of twelve occupational groups, the use pattern, and the exposure information of non-human hygiene disinfectant and insecticide products two product groups in occupational work places. Furthermore, we investigated valuable exposure factors, e.g., the patterns of use, exposure routes, and quantifying potential hazardous chemicals intake, on biocidal active ingredients. of occupational users by product use that are useful to conduct accurate exposure assessments. We focused on biocidal active ingredients exposure from products used by twelve occupational worker groups. The 685 non-human hygiene disinfectants and 763 insecticides identified contained a 152 and 97 different active ingredient chemicals. Toxicity values and clinical health effects about total twelve ingredient chemicals were determined through a brief overview of toxicity studies aimed at estimating human health risks. To estimate actual exposure amounts divided by twelve occupational groups, the time spent to apply the products was investigated from the beginning to end of the product use. This study investigated the exposure assessment of occupational exposure to biocidal products used in work occupational places, multi-use facilities, and general facilities. Furthermore, this study provides valuable information on the occupational exposure that may be useful to conduct accurate exposure assessment and to manage products used for quarantine in general facilities. The process of assessing exposure to products used in workplaces determine the patterns of use, exposure routes, and quantifying potential hazardous chemicals intake.

Q2) 2.1 interview-survey study: did the author receive any IRB from the Research Ethics Committee for interview surveys, online and telephone surveys? Since the authors surveyed workers, this paper should receive ethics approval from the Research Ethics Committee.

A2) Thank you for your comment.

The National Institute of Environmental Research (Government Institute) that I am working have ‘Standard Procedure for Institutional Review Board of National Institute of Environmental Research’ according to Bioethic Committee Regulation. I received IRB deliberation. The National Institute of Environmental Research and pre-review IRB determined that this study deliberation exemption, because this study does not collect or record personal information in accordance with the Personal Information Protection Act with an unspecific number of subject, etc. So, formal review board was not formed.

Q3) 2.4. Biocidal products and ingredients survey study: why is it important to know how much biocidal substances are manufactured and imported in Korea? The authors simply state that the total tonnage in Table 5 only, and no interpretation of the results can be found in the text. If this investigation is necessary, the results will need to be interpreted. In order for the author to calculate the amount used is calculated as manufactured volume + import volume – export volume. The authors should additionally calculated the export volume to know the distribution of chemicals.

A3) Thank you for your comment. I revised text in result part and interpretation.

3.4. Identification of active ingredient chemicals in biocidal products

Furthermore, in order to estimate potential exposure by grasping the amount to active ingredient chemicals distributed in South Korea, a total tonnage of manufacturing and importing biocidal products per year in Korea was surveyed (Table 5). The 685 non-human hygiene disinfectants and 763 insecticides identified in the product survey contained a 152 and 97 different active ingredient chemicals, including sodium hypochlorite, ethanol, hypochlorous acid, hydrogen peroxide, and didecyldimethylammonium chloride in disinfectants and tetramethrin, d-phenothrin, deltamethrin, and hydramethylnon in insecticides. Among the 685 disinfectants included in the subset, 108 products contained sodium hypochlorite and 83 products contained ethanol as the predominant active ingredient. Based on the results of distributed number of retail products, and total tonnage per year of manufactured and imported ingredients, several ingredient chemicals such as sodium hypochlorite, chlorine dioxide, and temephos were used in retail products in South Korean market.

Q4) 2.4. Biocidal products and ingredients survey study: the biocidal products and ingredients used were investigated through the interview survey. However, why did the authors research the same questions (products and active ingredients) from 459 companies? Are the 459 companies included in all 12 occupational categories in the interview survey to help readers understand, the authors need to write down the relationship between the interview survey and the online/telephone survey.

A4) Thank you for your comment. I revised text. Searching/survey from 459 companies was for obtain the information about products and their ingredient chemicals used. Interview survey was from 2432 respondents who were having occupations belong to twelve occupational groups.

In text, I revised these sentences.

3. Results

3.1. Characteristics of occupational consumer

As a second step, the interview-survey approach obtained the database about several sub-categories of twelve occupational groups, biocidal products use pattern in occupational places, and exposure information of two biocidal product groups. A total 2432 respondents, who were having occupations belonging to twelve occupational groups, completed the formal questionnaire during interview-survey. On the basis of survey results, the survey respondents were using various biocidal products in occupational places and were regularly using them. The sub-occupational distribution results of 2432 respondents (e.g., livestock occupation-livestock breeder, dairy worker, and livestock industry related machinery operation employee) were summarized in Table 2.

3.4. Identification of active ingredient chemicals in biocidal products

The searching for biocidal product available in retail markets identified 685 non-human hygiene disinfectants and 763 insecticides that offered substantial diversity in product purpose and active ingredient chemicals. In addition, we categorized a subset of searched products divided by the purpose for non-human hygiene disinfectants and the target insect intended for elimination or control from more in-depth analysis. Based on the results of the interview survey to the manufacturers and importers, 152 active ingredient chemicals used in non-human hygiene disinfectants, 97 active ingredient insecticides, and mixing ratio of these chemicals in products were identified. The purpose for insecticides was often defined based on the specific target insect intended for elimination or control. Common target insects included mosquitos, cockroaches, and house flies. Furthermore, in order to estimate potential exposure by grasping the amount to active ingredient chemicals distributed in South Korea, a total tonnage of manufacturing and importing biocidal products per year in Korea was surveyed (Table 5).

Q5) 2.5. Toxicological information for biocidal active ingredient chemicals: why the authors only investigated chronic toxicity information? Among the toxicity information, carcinogenicity, reproductive toxicities, and germ cell mutagenicity are particularly harmful substances. Have the authors considered investigating this toxicity information? If not, the authors should consider adding the following toxicity information.

A5) Thank you for your comment. I revised text about this point.

2.6. Toxicological information for biocidal active ingredient chemicals

The toxicological health effects and reference toxicity values (i.e., chronic NOAECs and NOAELs) of active ingredient chemicals were investigated. A occupational consumer exposure assessment was carried out according to the guidance from the information requirement and chemical safety assessment which was described as an efficient, step-wise, and iterative procedure (e.g., characterize the substance, determine the scope of exposure assessment, build/retrieve the contributing use scenario, estimate the event exposure, and carry out risk characterization). The target routes of exposure were considered to be inhalation and dermal route according to usage purpose and application types of products. An evaluation of the toxicological data was carried out in relation to the respiratory and irritant effects of long-term exposure to the ingredients under investigation (ECHA, 2016). Official toxicology reports and studies (i.e., United State Environmental Protection Agent (U.S. EPA) documents, United State California Environmental Protection Agent (U.S. California EPA) documents, United State Registration Eligibility Decision (U.S. RED) report, and the European Union European Chemical Agency (EU ECHA) Dossier) were used for each chemical in order investigate its toxicity value in the various products used. According to the toxicity value and health effects, dose rate and toxicokinetic information should be considered. In particular, we derived reference toxicity values based on chronic NOAELs (no observed adverse effect levels). If the value of chronic NOAELs could not be derived, differences in toxicological values (i.e., LOAELs (lowest observed adverse effect levels) and NOAELs) were taken into consideration.

Reviewer 2 Report

The authors aim to develop a representative database on exposure factors for biocidal products by occupational consumers and to evaluate toxicity characteristics of active ingredient chemicals used in these products. This is an engaging article and useful to increase our knowledge of the issue.

The title reports the key features of the paper encouraging the reader to read more. The abstract must be improved, the authors should be report the main results and conclusions. The introduction is quite clear what is already known about this topic. The aim of the paper is clearly justified but should be better outlined. The presentation of results is a little bit confusing in some points, and the discussion and conclusions must be improved. Addressing these issues will make this interesting paper eligible for the publication.

Some comments:

Materials and methods:

The study methods are not clear: who was the questionnaire addressed to? how were the subjects to be interviewed selected? what is the percentage of respondents out of the total?

The paragraph “Occupational use information of biocidal products” is not clear.

The authors should write a separate paragraph regarding statistical analysis.

Results:

The overall results section should be improved, it is not clear and in some points it is repetitive. The two interview schemes should be added as supplementary material.

Some points:

how many respondents are there for each occupational consume?

where are the statistical analysis presented?

since the use of different disinfectants with different active chemical ingredients, does it make sense to present the exposure amount mean per day or per use without distinction on the basis of the used biocidal product (e.g. Table 4)?

Check in the results section the reference to Table 6.

Discussion:

The authors should reorganize the discussion by making it less dispersive and discussing some critical points of their study. The first part seems be a part of introduction section. The authors should discuss their results. They should start this section with their main and innovative points of their results comparing with the previous literature. They should discuss the limitations and the strengths of their study. For example, their data is obtained through interview-survey and not measured.

Conclusions:

The authors should write the conclusions taking into account their initial aims and the limitations of their study design. In addition, further studies should be done to strengthen their results.

Author Response

Thank you very much for your comments and great suggestions. Your detailed and very valuable comments and recommendations improved the quality and presentation of this manuscript. I appreciate your time and effort. I answered to your questions and comments item by item. Most of your suggestions were reflected in the revision, and were detailed in the following.

Q1) the title reports the key features of the paper encouraging the reader to read more.

A1) Thank you for your comment. I revised title.

A Potential Health Risks to Biocidal Active Chemicals via Biocide Products Exposed by Non-professional Occupational User

Q2) the abstract must be improved, the authors should be report the main results and conclusions.

A2) Thank you for your comment. I revised abstract.

Abstract: Biocidal active chemicals have potential health risks associated from exposure to retail biocide products pesticides and disinfectants such as personal disinfectants for COVID-19. Reliable exposure assessment was investigated to understand the exposure pattern of biocidal product used by occupational workers in occupational place, multi-use facilities, and general facilities. Exposure to hazardous biocidal chemicals via products used by occupational workers in occupational place, multi-use facilities, and general facilities may play a role in health risk of occupational workers and general public caused by use of ingredients. The interview-survey approach was conducted to obtain the database about several sub-categories of twelve occupational groups, the use pattern, and the exposure information of non-human hygiene disinfectant and insecticide products two product groups in occupational work places. Furthermore, we investigated valuable exposure factors, e.g., the patterns of use, exposure routes, and quantifying potential hazardous chemicals intake, on biocidal active ingredients. of occupational users by product use that are useful to conduct accurate exposure assessments. We focused on biocidal active ingredients exposure from products used by twelve occupational worker groups. The 685 non-human hygiene disinfectants and 763 insecticides identified contained a 152 and 97 different active ingredient chemicals. Toxicity values and clinical health effects about total twelve ingredient chemicals were determined through a brief overview of toxicity studies aimed at estimating human health risks. To estimate actual exposure amounts divided by twelve occupational groups, the time spent to apply the products was investigated from the beginning to end of the product use. This study investigated the exposure assessment of occupational exposure to biocidal products used in work occupational places, multi-use facilities, and general facilities. Furthermore, this study provides valuable information on the occupational exposure that may be useful to conduct accurate exposure assessment and to manage products used for quarantine in general facilities. The process of assessing exposure to products used in workplaces determine the patterns of use, exposure routes, and quantifying potential hazardous chemicals intake.

Q3) the introduction is quite clear what is already known about this topic. The aim of the paper is clearly justified but should be better outlined. The presentation of results is a little bit confusing in some points and the discussion and conclusions must be improved. Addressing these issues will make this interesting paper eligible for the publication.

A3) Thank you for your very valuable comment. According to your comment, I revised text.

4. Discussion

The aim of this study was to create a national exposure factor database for use in exposure and risk assessments of biocidal products in terms of human health. The study mainly showed that the biocidal products use in prevalent occupational workplace, multi-use facilities, and general facilities could cause the potential exposure of their active ingredient chemicals to occupational consumers and general public.

The specific pattern of use data requirements for different biocidal products are purpose of product (physical properties, where used, description of tasks, target organisms), use environment (pattern of control, use pattern), loading phase (task, frequency per task, duration of task, quantity used per task), application phase, post-application phase, and others. The essential information of use pattern required to derive exposure scenarios, which are then evaluated to derive quantitative exposure estimates [16-17]. For the survey study used in this study, respondents had to declare the frequency, duration, and amount of product used per application of biocidal products, given several options (SI 1). However, surveys have their limitations. A lack of accurate memory regarding the questions posed might impact the quality of the data; in person interview surveys might yield data of better quality [21-22]. For health risk assessments concerning volatile ingredients, vapor pressure of ingredient chemicals should be considered. In order to measure the accurate concentration of volatile ingredients in the air of occupational environment, residual volatile ingredients emitted into indoor air should be measured. However, in this study, we could not measure these amounts and concentrations in indoor air. This approach might be useful in establishing guidelines of exposure assessment for occupational consumers. To protect consumers and public users from several hazardous ingredient chemicals, more comprehensive exposure estimation and assumption are needed.

5. Conclusions

This study investigated a fundamental approach to assess the occupational exposure to biocidal active ingredients by using products in occupational places (workplaces), multi-use facilities, and general facilities. The process of assessing exposure to biocidal products used in workplaces requires determining the patterns of use (exposure factors), identifying the exposure population (occupational consumers), establishing exposure routes (inhalation and dermal exposure), and quantifying potential ingredient chemicals intake. This study determined the recent exposure factors using a interview survey of over 2400 occupational consumers using biocidal products in occupational place, multi-use facilities, and general facilities. Estimating occupational exposure to biocidal active ingredient chemicals via using products is a fundamental element of the health risk assessment process. Additionally, we evaluated the exposure amount of biocidal product to consumers and toxicological characteristics of ingredient chemicals, considering the characteristics of occupational characteristics and biocidal products categories. As a further study, health risk assessment study of active ingredients to occupational consumers and public users of multi-use facilities and general facilities.

Q4) materials and method: the study methods are not clear: who was the questionnaire addressed to? How were the subjects to be interviewed selected? What is the percentage of respondents out of the total?

A4) Thank you for your comment. As your comment, I revised text.

2.1. Interview-survey study

The biocidal product consumers were categorized as the general public consumers (household users) and occupational consumers. Occupational consumers were defined as the consumer of occupational groups who use biocidal products in workplaces and public, multi-use facilities with high frequency because of their occupational characteristics. who may be exposed to chemicals by using products that could be purchased from retail markets. In order to characterize occupational consumers using biocidal product, the Korean Standard Classification of Occupation was referenced. Due to the characteristics of the occupation, the twelve occupational groups using a lot of biocidal products were selected. An interview-survey study was conducted to elucidate which biocidal products were commonly used in occupational places (workplaces) across all cities and provinces in Korea by occupational consumers. Interview-survey was carried out a Korean survey company (K-STAT Research Ltd) that we hired. Survey company has participants fool in all provinces, cities in Korea. Participants were selected as follow 1) workers having occupation belonging to the twelve occupational groups, and 2) workers using biocidal products in workplace because of occupational characteristics. If participants agreed to take the interview-survey among them answered that have experience using target biocidal products, interviewer conducted an individual interview. On the basis of the interview results, the survey company conducted a market searching to identify and characterize the retail biocidal products and ingredient chemicals that participants used.

Q5) the paragraph ‘occupational use information of biocidal products’ is not clear.

A5) Thank you for your comment. I revised this sentence.

2.3. Occupational use information Use pattern of biocidal products

Q6) the authors should write a separate regarding statistical analysis. Where are the statistical analysis presented?

A6) Thank you for your comment. I revised text.

2.4. Statistical analysis

All statistical analyses were conducted using SPSS version 22.0 differences with a P value of less than 0.05 were considered to be statistically significant unless noted otherwise. As a statistical analysis method to verify the difference between groups, T-test or one-way ANOVA was carried out.

Q7) results: the overall results section should be improved, it is not clear and in some points it is repetitive. The two interview schemes should be added as supplementary material.

A7) Thank you for your comment. According to your comment, I revised some part of results and I added “Supporting Information (S1) 1. Questionnaire of interview-survey to occupational workers used disinfectant for kitchen”.

But, I did not show the question scheme of manufacturing and importing companies, because Survey company simply asked, only what kinds of chemicals for biocidal prodcuts production were used in your company? And how many tonnage per year were those chemicals produed and imported?

Q8) how many respondents are there for each occupational consumer?

A8) Thank you for your comment. As your comment, I inserted numbers of each occupational consumers in Table 3.

Q9) since the use of different disinfectants with different active chemical ingredients, does it make sense to present the exposure amount mean per day or per use without distinction on the basis of the used biocidal product (e.g. Table 4)?

A9) Thank you for your comment. About exposure & risk assessment study, I followed ECHA guidance and RIVM document (reference 16 ~ 18). These factors (use amount in Table 4) means “the worst-case use scenario and estimation”.

1 step : identification of hazardous substances

2 step : quantifying of hazardous substances in target products

3 step : selecting toxicological information of target substances & hazardous identification

4 step : exposure assessment (determining exposure factors to studied subjective)

Next step is risk assessment, this study is under studying.

Evaluation of chemical or products, different assessment methods are needed. I used the worst-case scenario for exposure. In this study, I investigated exposure information and exposure factors. As a further study, I will do exposure assessment & health risk assessment study divided single chemicals in single products, single chemicals in multi products (cumulative exposure), multi chemicals in single products (combined exposure).

Q10) check in the results section the reference in Table 6.

A10) Thank you for your valuable comment. I revised text.

3.4. Identification of active ingredient chemicals in biocidal products

~ The 685 non-human hygiene disinfectants and 763 insecticides identified in the product survey contained a 152 and 97 different active ingredient chemicals, including sodium hypochlorite, ethanol, hypochlorous acid, hydrogen peroxide, and didecyldimethylammonium chloride in disinfectants and tetramethrin, d-phenothrin, deltamethrin, and hydramethylnon in insecticides. Among the 685 disinfectants included in the subset, 108 products contained sodium hypochlorite and 83 products contained ethanol as the predominant active ingredient. Based on the results of distributed number of retail products, and total tonnage per year of manufactured and imported ingredients, several ingredient chemicals such as sodium hypochlorite, chlorine dioxide, and temephos were used in retail products in South Korean market. Table 6 summarizes the ten most common active ingredients used in non-human hygiene disinfectants and insecticides, including their prevalence, maximum concentrations, and total tonnage of manufacturing and importing biocidal products per year. ~

Q11) discussion : the authors should recognize the discussion by making it less dispersive and discussing some critical points of their study. The first part seems be a part of introduction section. The authors should discuss their results. They should start this section with their main and innovative points of their results comparing with the previous literature. They should discuss the limitations and the strengths of their study. For example, their data is obtained through interview survey and not measured.

A11) Thank you for your comment. I revised discussion part.

The aim of this study was to create a national exposure factor database for use in exposure and risk assessments of biocidal products in terms of human health. The study mainly showed that the biocidal products use in prevalent occupational workplace, multi-use facilities, and general facilities could cause the potential exposure of their active ingredient chemicals to occupational consumers and general public. ~ For the survey study used in this study, respondents had to declare the frequency, duration, and amount of product used per application of biocidal products, given several options (SI 1). However, surveys have their limitations. A lack of accurate memory regarding the questions posed might impact the quality of the data; in person interview surveys might yield data of better quality [21-22]. For health risk assessments concerning volatile ingredients, vapor pressure of ingredient chemicals should be considered. In order to measure the accurate concentration of volatile ingredients in the air of occupational environment, residual volatile ingredients emitted into indoor air should be measured. However, in this study, we could not measure these amounts and concentrations in indoor air. This approach might be useful in establishing guidelines of exposure assessment for occupational consumers. To protect consumers and public users from several hazardous ingredient chemicals, more comprehensive exposure estimation and assumption are needed.

Q12) conclusions: the authors should write the conclusions taking into account their initial aims and the limitations of their study design. In addition, further studies should be done to strengthen their results.

A12) Thank you for your comment. As your comment, I revised conclusion.

5. Conclusions

This study investigated a fundamental approach to assess the occupational exposure to biocidal active ingredients by using products in occupational places (workplaces), multi-use facilities, and general facilities. The process of assessing exposure to biocidal products used in workplaces requires determining the patterns of use (exposure factors), identifying the exposure population (occupational consumers), establishing exposure routes (inhalation and dermal exposure), and quantifying potential ingredient chemicals intake. This study determined the recent exposure factors using a interview survey of over 2400 occupational consumers using biocidal products in occupational place, multi-use facilities, and general facilities. Estimating occupational exposure to biocidal active ingredient chemicals via using products is a fundamental element of the health risk assessment process. Additionally, we evaluated the exposure amount of biocidal product to consumers and toxicological characteristics of ingredient chemicals, considering the characteristics of occupational characteristics and biocidal products categories. As a further study, health risk assessment study of active ingredients to occupational consumers and public users of multi-use facilities and general facilities.

Round 2

Reviewer 1 Report

I accept for publication after revising two additional remarks below.

1. The authors should translate the foreign language of supplementary materials into English.

2. Following the author's guideline, the authors should write down the information about the supplementary material below the conclusions.

Author Response

Thank you very much for your comments and great suggestions (second revision). Your detailed and very valuable comments and recommendations improved the quality and presentation of this manuscript . Most of your suggestions were reflected in the revision, and were detailed in the following.

Q 1 ) the authors should translate the foreign language of supplementary materials into English
A1) Thank you for your comment. I revised supplementary information according to your comments.

Q 2 ) following the author’s guideline, the authors should write down the information about the supplementary information below the conclusions.
A2) Thank you for your comment. I revised text. Associated Content: Questionnaire of interview-survey is presented as supporting information (SI 1).

Reviewer 2 Report

The research topic of this manuscript is of interest and highly relevant to public need. Most of the reviewers' comments have been well addressed. I would suggest acceptance of the manuscript after minor revisions.

My comments are listed below.

Lines 124-127: this paragraph should be moved to the end of the materials and methods.

Line 174: what is the percentage of respondents out of the total?

Table 3:

  • the formula for exposed amount should be reported in table 4 and not in table 3;
  • why the sum of the respondents divided by occupational consumer does not correspond to the data reported in the text of 2432?

Line 315: this sentence could start with “This study aimed to…”

Lines 408-411: arrange the English form in order to clarify the sentence.

Supplementary information should be reported in English.

Author Response

Thank you very much for your comments and great suggestions (second revision). I appreciate your time and effort. I answered to your questions and comments item by item. Most of your suggestions were reflected in the revision, and were detailed in the following.

Q1) line 124 127: this paragraph should be moved to the end of the materials and methods.
A1) Thank you for your comment. I revised this paragraph. 2.6. Statistical analysis All statistical analyses were conducted using SPSS version 22.0 differences with a P value of less than 0.05 were considered to be statistically significant unless noted otherwise. As a statistical analysis method to verify the difference between groups, T-test or one-way ANOVA was carried out.

Q2) line 174 : what is the percentage of respondents out of the total?
A2) Thank you for your comment. I revised text in order to explain this question. 3.1. Characteristics of occupational consumer ~ As a second step, the interview survey approach obtained the database about several sub categories of twelve occupational groups, biocidal products use pattern in occupational places, and exposure information of two biocidal product groups. A to tal 2432 respondents, who were having occupations belonging to twelve occupational groups, completed the formal questionnaire during interview survey. A contact for interview survey (purpose number of respondents was over 2000cases) were continuously proce ssed to participants until reaching purpose cases. On the basis of survey results, the survey respondents were using various biocidal products in occupational places and were regularly using them. The sub occupational distribution results of 2432 responden ts ( e.g., livestock occupation livestock breeder, dairy worker, and livestock industry related machinery operation employee) were summarized in Table 2.

Q3) table 3: the formular for exposed amount should be reported in table 4 and not in table 3.
A3) Thank you for your very valuable comment. According to your comment, I revised text. 
c: exposed amount per day = use frequency(use/day) × use amount(g/use).: exposed amount per day = use frequency(use/day) × use amount(g/use).

Q4) why the sum of the respondents divided by occupational consumer does not
correspondent to the data reported in the text of 2432?
A4) Thank you for your comment. As your comment, I revised number error. Children care business (n = 200)

Q5) line 315: this sentence could start with “this study aimed to ~”
A5) Thank you for your comment. I revised this sentence. 4. Discussion This study aimed to create a national exposure factor database for use in exposure and risk

Q6) line 408 411: arrange the English form in order to clarify the sentence.
A6) Thank you for your comment. I revised text. . Furthermore, we calculated the exposure amount of biocidal product used by occupational consumers. Exposure amount of products could be used for estimating exposure amount of their ingredient chemicals to occupational consumers. Additionally, toxicological characteristics of ingredient chemicals were evaluated considering the characteristics of occupational characteristics and biocidal products categories. As a further study, health risk assessment study of active ingredients to occupational consumers, and public users of multi-use facilities and general facilities as a second exposure estimation were processed.

Q7) supplementary information should be reported in English.
A7) Thank you for your comment. According to your comment, I revised supplementary
information
